# Application of Advanced Technologies—Nanotechnology, Genomics Technology, and 3D Printing Technology—In Precision Anesthesia: A Comprehensive Narrative Review

**DOI:** 10.3390/pharmaceutics15092289

**Published:** 2023-09-06

**Authors:** Shiyao Gu, Qingyong Luo, Cen Wen, Yu Zhang, Li Liu, Liu Liu, Su Liu, Chunhua Chen, Qian Lei, Si Zeng

**Affiliations:** 1Department of Anesthesiology, Sichuan Provincial People’s Hospital, School of Medicine, University of Electronic Science and Technology of China, Chengdu 610072, China; 2School of Medical and Life Sciences, Chengdu University of Traditional Chinese Medicine, Chengdu 610072, China; 3Department of Anesthesiology, Affiliated Hospital of Xuzhou Medical University, Xuzhou 221000, China; 4Department of Anatomy and Embryology, School of Basic Medical Sciences, Peking University Health Science Center, Beijing 100191, China

**Keywords:** anesthesia, algology, 3D printing, nanotechnology, genomics, precision medicine, anesthetic safety

## Abstract

There has been increasing interest and rapid developments in precision medicine, which is a new medical concept and model based on individualized medicine with the joint application of genomics, bioinformatics engineering, and big data science. By applying numerous emerging medical frontier technologies, precision medicine could allow individualized and precise treatment for specific diseases and patients. This article reviews the application and progress of advanced technologies in the anesthesiology field, in which nanotechnology and genomics can provide more personalized anesthesia protocols, while 3D printing can yield more patient-friendly anesthesia supplies and technical training materials to improve the accuracy and efficiency of decision-making in anesthesiology. The objective of this manuscript is to analyze the recent scientific evidence on the application of nanotechnology in anesthesiology. It specifically focuses on nanomedicine, precision medicine, and clinical anesthesia. In addition, it also includes genomics and 3D printing. By studying the current research and advancements in these advanced technologies, this review aims to provide a deeper understanding of the potential impact of these advanced technologies on improving anesthesia techniques, personalized pain management, and advancing precision medicine in the field of anesthesia.

## 1. Introduction

Precision medicine applies individualized clinical or genetic information to inform decisions regarding disease prevention, diagnosis, and treatment. It is also termed individualized medicine, individualized genomics, or genomic medicine [1]. Precision medicine, a rapidly evolving field, has demonstrated its potential in various biomedical areas and addresses major public health concerns. In cancer research, precision medicine has revolutionized treatment approaches by tailoring therapies based on a patient’s specific genetic mutations or tumor characteristics [2]. Similarly, in neurodegenerative diseases such as Alzheimer’s disease, precision medicine holds the potential for systemic interventions [3]. Precision medicine has also contributed to advancements in managing heart diseases [4], optimizing anesthesia techniques [5], and even studying precision treatments for diseases like COVID-19 [6]. It seeks to facilitate the correct selection and precise application of appropriate treatment methods for each patient in order to minimize medical damage and costs as well as maximize patient benefits. The early focus of personalized medicine was on genetic testing [7].

The US proposed the Precision Medicine Initiative, which seeks to promote the use of existing and new genomic databases to improve diagnosis and treatment [8]. Although precision medicine emphasizes the use of genetic test results to inform diagnosis and treatment, integrating both advanced and traditional medical approaches is another major objective of precision medicine. Traditional clinical anesthesiology offers few auxiliary tools; rather, it relies more on the knowledge system, proficiency, and clinical experience of anesthesiologists [9]. Accordingly, given the resulting subjectivity and limited human ability, this approach often involves intraoperative discomfort and adverse postoperative effects. These consequences may include challenges in clinical pain management, as the assessment and management of pain often depend on the evaluator’s experience in recognizing pain-related clinical signs. Additionally, differences in anesthesia plans among different anesthesiologists for individual patients may lead to increased risks of adverse reactions during or after surgery [10].

With the advent of new technologies such as combined anesthesia, visual laryngoscopy, and nerve blocks under ultrasonography, there has been continuous improvement in clinical anesthesia techniques, increased safety and stability of anesthesia protocols, and a corresponding decrease in postoperative complications [11,12,13]. Further, emerging medical fields such as nanotechnology, genomics technology, and 3D printing technology have opened up new avenues in targeted precision therapy [14] and personalized, customized anesthesia treatment plans [15]. This study has searched the PubMed database using anesthesia, 3D printing, nanotechnology, genomics, and precision medicine as search terms for articles published between 2000 and 2023. The retrieved articles include Clinical Trials, Randomized Controlled Trials, and Reviews. Subsequently, the articles have been classified based on their abstract content. This article reviews current developments in nanotechnology, genomics technology, and 3D printing technology with respect to clinical anesthesia and explores the possibility of achieving precision anesthesia. By studying the current research and advancements in these advanced technologies, this review aims to provide a deeper understanding of the potential impact of these advanced technologies on improving anesthesia techniques, personalized pain management, and advancing precision medicine in the field of anesthesia.

## 2. Application of Nanotechnology in the Anesthesia Field

### 2.1. Anesthesia and Nanotechnology

Nanotechnology has advanced into almost all fields of science, including physics, materials science, chemistry, biology, computer science, and engineering [16]. Nanotechnology is defined as the control of substances sized 1–100 nm [17]. Nanomedicine involves unique bio-interactive surface modifications and disease-targeting drug-loaded encapsulated nanoparticles [18]. As shown in Figure 1, several nanocarriers have been recently established in the field of medicine, including nanoparticles, nanoliposomes, liposomes, micelles, solid lipid particles, surfactant vesicles, and other nanodevices. The drug slow-release effect of these medications has shown significant advantages [19]. Nanotechnology has been shown to allow for more effective, precise, and safer treatment of cancer [14,20]. Further, nanotechnology is crucially contributing to drug delivery system therapy [21], imaging [22], and immunotherapy [23], given its unique particle-size characteristics. Taken together, nanotechnology is being extensively applied in clinical medicine. Moreover, nanomedicine is being utilized in various sub-disciplines to achieve precision and personalized medicine.

In clinical anesthesiology, there is a crucial need to maintain stable vital signs and resolve emergencies during the perioperative period, including maintaining an appropriate depth of anesthesia, sedation, amnesia and analgesia, muscle relaxation, neurovegetative protection, and reducing stress reactions [36,37]. With technological advancements, anesthesiologists should not only master basic anesthesia operations but also take advantage of the convenience allowed by advanced science and technology. Accordingly, nanomedicine may allow advancement and optimization of the practice of clinical anesthesiology in perioperative settings. Anesthesiologists can apply nanotechnology to guide routine clinical anesthesia and pain control for an improved perioperative patient experience, reduced risk of perioperative-related complications, and progression toward precision medicine [38]. Inhalation nanoparticles allow enhanced saturation solubility, rapid dissolution kinetics, and high drug concentrations that rapidly reach the absorption site [39]. On the other hand, local anesthetic nanoparticles have the potential to reduce drug toxicity and prolong the duration of drug action [27]. The mixed nanofilm patch formulated with lidocaine and prilocaine demonstrates enhanced permeability and a prolonged anesthetic effect [24]. As shown in Table 1 and Figure 2, nanoparticles provide safe and optimal conditions, including nanotechnology-controlled inhalation and local analgesia, precise anesthetic delivery, and perioperative pain control.

#### 2.1.1. Nanotechnology in Inhalation Anesthesia

Intravenous drug delivery is a widely used therapeutic method in clinical settings. However, it has several disadvantages, including wide drug dispersion [40], numerous systemic reactions, poor precision, and poor drug solubility, which limit the scope of drug administration and clinical treatment effects because it depends entirely on the mechanisms of biotransformation and elimination of the organism so that the drug can exert its therapeutic effect. Therefore, this itself can prolong the adverse effects. Nanoparticles have advantages in pulmonary drug delivery given their small particle size and fast absorption. In vitro and in vivo studies have investigated the performance of nanopolymer forms in pressurized metered-dose inhalers, nebulizers, and dry powder respirator applications, as well as the deposition, dissolution, uptake, and clearance pathways of nanoparticles in the respiratory tract [41]. All the studied nanoparticles have demonstrated excellent performance and have clearly demonstrated that inhaled nanoscale formulation particles can enhance solubility [39], improve bioavailability and patient compliance [42], enhance therapeutic efficacy, and reduce the possibility of drug dosage [43]. Furthermore, concerns have been raised regarding the increased toxicity associated with the delivery of non-therapeutic molecules through inhaled nanoparticles. The suitability of inhaled nanoparticles for drug molecules [43] and their potential applications in different therapeutic areas warrant further research to unveil their full potential. Although there are currently not many clinical applications of nanotechnology in inhalation drug delivery, establishing a novel drug administration route using lung inhalation nanoparticles could allow for an innovative approach. Salama et al. [44] formulated novel dispersions for the volatile inhalation of the anesthetic sevoflurane with a diameter of 250 nm. They customized the flow device to provide controlled, sustained, and adjustable release of the anesthetic across a clinically usable time frame. The simplicity, inexpensiveness, portability, and controllability of the device allowed easier use than conventional delivery systems for anesthetic gases and could inform further development of nanotechnology for inhalation anesthetic formulations. Moreover, a clinical trial using six healthy male volunteers prepared atropine sulfate as a dry powder inhaler for absorption through the lungs. The in vitro and in vivo experiment results showed that this preparation allowed easy administration, rapid onset of action, painlessness, improved stability performance, high bioavailability, and sustained action of the intestinally absorbed drug [45]. Although this previous study only used atropine dry powder inhalation formulation for organophosphorus poisoning detoxification, further advances in nanomedicine could expand the clinical utility of the nano-inhalation formulation of the drug [45]. Accordingly, further studies on inhaled nanoformulations could expand their clinical applications. The application of nanoparticles has managed to expand the administration of inhalation anesthetics, as shown in a study conducted in dogs. In this study, dogs were given either inhaled isoflurane (Iso-I) or intravenous 15% isoflurane-loaded lipid nanoemulsion (Iso-nano) for maintenance of anesthesia. The results of this study show that with the use of this innovative technology, it is possible to maintain general anesthesia in the animal biological model, which would possibly have a benefit in reducing adverse effects. However, it did not reduce the amount of isoflurane required for maintaining general anesthesia and led to significant hypotension and non-respiratory acidosis [46]. Indeed, these findings raise the question of whether inhaled sevoflurane nanoparticles may possess similar or even superior characteristics. However, further research is needed to investigate these possibilities. Meanwhile, in a study investigating the therapeutic targeting of the ischemic region in the brain, a system was developed using a Fas ligand-conjugated reduced graphene oxide (rGO) system loaded with sevoflurane (SF). It was observed that the specific nanoparticles selectively accumulated in the ischemic region of the mouse model being tested [47]. Indeed, there are still many mysteries waiting to be unraveled in the field of inhaled nanomedicines.

#### 2.1.2. Nanotechnology in Local Anesthesia

Intravenous regional anesthesia (IVRA) is often used in short and minor extremity surgeries given its ease and simplicity. However, it is limited by local anesthetic systemic toxicity reactions, the placement of tourniquets, and poor anesthetic outcomes, which result in a poor patient experience and place the anesthesiologist in a relatively passive position [48,49,50]. Weldon et al. [25] investigated the effects of local anesthesia with bupivacaine nanoencapsulation in a rat intracaudal vein local anesthesia model. Specifically, they caudally injected free bupivacaine, a 15 nm micellar bupivacaine formulation (M-Bup), and a 100 nm liposomal bupivacaine formulation into the rat model. They performed in vitro release assays and cytotoxicity assays, as well as measuring the duration of intravenous local anesthetic effects. In mice, tail analgesia lasted for 2.0 ± 0.6 h with 0.5% free bupivacaine. In contrast, the duration of tail analgesia was more than doubled in the M-Bup group with 0.1% bupivacaine, lasting for 4.5 ± 0.5 h, despite the bupivacaine dosage being one-fifth of the original [25]. Fluorescence and pharmacokinetic studies in whole animals and tissues have shown that poor performance is attributed to low tissue adsorption [25]. The surface polyethylene glycol (PEG) coating of nanoparticles is commonly employed to reduce interactions with serum proteins and cells [51,52], thereby increasing circulation time in the bloodstream and enhancing accumulation at the site of disease. M-Bup exhibited a longer residence time in the local vasculature. Further, there was sustained bupivacaine release, which significantly prolonged the duration of local intravenous anesthesia, with M-Bup being the most effective. Notably, M-Bup showed relatively reduced systemic drug distribution. Further clinical studies are warranted to explore the patient safety and anesthesia persistence of using M-Bup for IVRA.

**Table 1 pharmaceutics-15-02289-t001:** Application of Nano Agents in Local Anesthesia.

Local Anesthetic Drugs	Nanoparticle Type	Application Path	Test Method	Results	Duration of Efficacy
Bupivacaine	15 nm micellar bupivacaine formulation (M-Bup) and 100 nm liposomal bupivacaine formulation	Local anesthesia	Tail injection in rats [25]	It showed an extended residence time in the local vasculature, with M-Bup showing the most prominent effect; there was also a reduction in systemic drug distribution [25].	M-Bup provides 4.5 h of local anesthesia [25].
	Muhilamellar liposomes	Local anesthesia	Brachial plexus anesthesia in rabbits [28], intravenous drip in rabbits’ ears [29]	There was a prolonged effect of local anesthetics [28], with a significantly reduced drug toxicity to the central nervous system and heart [29].	In the BP-MLV group, the plasma concentration of bupivacaine was lower within the first 10 min (*p* < 0.05) and higher after 24 h (*p* < 0.05). The radiolabeling in the BP group decreased between 4 and 24 h, while in the BP-MLV group, it decreased between 1 and 2 days [28].
	Polymerized alginate nanoparticles [27,30], large multicapsular liposomes (Bupisome) encapsulated in Ca-alginate cross-linked hydrogels (Bupigel)	Local anesthesia	In vitro and in vivo testing in mice, physicochemical property determination [27], and subcutaneous injection in mice [30]	It has good stability, low cytotoxicity, and a strong intensity of action [27]. There was a prolonged duration of the analgesic effect [27,30], with Bupigel outperforming Bupisome.	BVC (bupivacaine) is completely released in the solution after 350 min (100%), while the complete release of BVC present in the nanoparticle takes a longer time [27].
	Large multivesicular vesicles	Local anesthesia	Healthy volunteers received subcutaneous injections [53]	Delayed elimination and prolonged redistribution of plasma results in prolonged pharmacodynamic effects [53].	The time to reach the maximum plasma concentration of the liposomal formulation increased by 7-fold (262 +/− 149 min vs. 37.5 +/− 16 min, *p* < 0.01) [53].
	Liposomal bupivacaine (LEB)	Local anesthesia	Intra-articular soft tissue injection in dogs [54]	Dogs administered with LEB are less likely to require rescue analgesia and receive lower doses of opioid medications compared to dogs administered with 0.5 BH [54].	In the LEB group, three dogs requiring rescue analgesia were identified at 8 h (*n* = 2) and 16 h (*n* = 1) post-extubation, based on a CSU-CAPS pain score ≥ 2. In the 0.5 BH group, among the 10 dogs requiring rescue analgesia, 7 dogs first exhibited these symptoms within 6 (*n* = 4) to 8 (*n* = 3) hours post-extubation [54].
	Liposomal suspension of bupivacaine	Local anesthesia	Sciatic nerve blockade in dogs [55]	The blockade characteristics of bupivacaine liposomal suspension are effective and long-lasting [55].	In the treatment of 10 cases with bupivacaine with dexmedetomidine (BUP-DEX), all functions completely disappeared at 6 h. In all cases, all functions recovered within 96 h and 24 h after administration of bupivacaine liposome suspension (BLS) and BUP-DEX, respectively [55].
	Microcapsules	Local anesthesia	Assessment of catheter microdialysis in healthy volunteers [56]	The extended-release properties of microcapsules allow a prolonged duration of anesthesia [56].	After injection of microcapsules, the concentration of bupivacaine increased within 24–34 h. After 96 h, 78% of the injection sites with microcapsules still had analgesic effects, significantly longer than the bupivacaine solution (*p* < 0.001) [56].
Prilocaine	Liposomes	Local anesthesia for oral cavity	Maxillary infiltration anesthesia in healthy volunteers [31]	Prilocaine does not seem to benefit from liposome encapsulation [31].	The median (and interquartile range) onset time for all formulations of gingival anesthesia was 2 (0) minutes, with no significant difference between them (*p* > 0.05) [31].
	Liposomes complexed with hydroxypropyl-β-cyclodextrin	Local anesthesia	In vivo assessment of anesthetic effects in guinea pigs [32]	The duration of the anesthetic effect was negatively correlated with the initial lag time of PRL hydrochloride in the core of aqueous vesicles [32].	Dual-loaded liposomes containing 2% of the total drug dose exhibited optimal therapeutic activity and were significantly superior to the corresponding 2% single-loaded vesicles. They not only showed the shortest onset time (100% blockade of reflexes at 5 min) but also the longest duration of anesthesia effect (100% blockade of reflexes at 35 min) [32].
Lidocaine	Liposomes	Surface local anesthesia	Skin test on the palmar side of the forearm of volunteers [26]	Lidocaine liposome anesthesia has a longer duration than regular preparation [26].	The average pain score of 5% liposomal lidocaine was higher than the non-liposomal 5% lidocaine formulation, but the difference reached statistical significance only at 15 min [26].
Lidocaine-Prilocaine	Hybrid Nano Film	Local anesthesia	Permeability test of porcine buccal mucosa, tail-flick test in mice [24]	It is more permeable and has a longer anesthetic effect; it is not cytotoxic to 3T3 and HACAT cell lines [24].	The obtained material showed a sustained release profile of LDC-PLC for over 8 h, and the permeability of pig buccal mucosa was nearly double that of the control group. Then, the in vivo efficacy of the PCT/NLC formulation was compared to biopolymer films and commercial drugs, demonstrating the longest anesthetic effect (>7 h) in mice through a tail flick test [24].
Mepivacaine	Liposomes	Oral local anesthesia	Oral injection [33], oral maxillary infiltration [34] in healthy volunteers	It extends the duration of anesthesia, reduces injection discomfort [33], and allows systemic absorption similar to that of vasoconstrictor local anesthesia [34].	Healthy volunteers experienced median ranges of induction latency (LP) (2–8 min), pulpal anesthesia (PA) (20–45 min), and soft tissue anesthesia (STA) (120–180 min) after infiltration anesthesia with the following formulations of lidocaine: MVC 2%EPI, MVC 2%LUV, MVC 3%LUV, and MVC 3% [33].
Ropivacaine	Liposomes	Oral local anesthesia	Maxillary infiltration in healthy volunteers [35]	Liposome formulations lack vasoconstrictors and may be a safer alternative [35].	They observed a maximum drug concentration (Tmax) of 50 (±14.1) minutes [35].
Tetracaine (TTC)	Polymeric nanoparticles (PLA NPs), solid lipid nanoparticles (SLNs), and nanostructured lipid carriers (NLCs)	Local anesthesia	In vitro and in vivo tests in mice [57]	Each system has its advantages, with TTC NLCs being the more promising system for long-term anesthesia [57].	Free TTC demonstrated complete permeation within 8 h, while TTC NLC showed lower permeation rates than TTC PLA NPs in the first 12 h but higher permeation rates than PLA NPs after 12 to 72 h. TTC SLNs provided the most effective in vitro permeation, with sustained efficiency lasting until the end of 72 h [57].

Similarly, other animal studies have demonstrated that liposome-related bupivacaine and proparacaine (PRC) allow a higher duration of anesthetic effects than common free drugs [32]. Specifically, compared with common free drugs, bupivacaine-related polymeric alginate nanoparticles have a higher intensity of anesthetic effects, a longer anesthesia duration, and lower cytotoxicity [27]. Furthermore, Ca-alginate cross-linked hydrogels show recommendable stability in storage [30]. Although most of the aforementioned reports are experimental animal studies, there have been recent clinical trials on the application of nanotechnology. Davidson et al. [53] performed successive subcutaneous injections of 20 mL of plain 0.5% bupivacaine and 20 mL of 2% liposomal bupivacaine in the vesicular vesicles of eight healthy volunteers and analyzed blood specimens from control-related nodes. They observed no between-group differences in the peak bupivacaine plasma concentrations or toxic responses, despite the total bupivacaine dose in the liposomal formulation being four times that in the plain formulation. Kopacz et al. [56] reported that subcutaneous injections of bupivacaine in an aqueous solution at thrice the dose that microcapsules containing bupivacaine and dexamethasone produced had a relatively shorter duration of dermal anesthesia and analgesia. In these studies, the small doses of local anesthetic nanoformulations compensated for their short anesthesia duration. Contrastingly, high-dose nanoformulations compensated for their poor safety profile, which demonstrated their safety and persistent effects.

Local anesthetics are often used in combination with epinephrine for clinical anesthesia to limit the adverse effects of widespread diffusion absorption. However, epinephrine has vasoconstrictive effects. Moreover, its improper application presents a risk of ischemia and distal limb necrosis [58] and may have severe cardiac effects. Animal studies have shown that, compared with solutions of plain bupivacaine or bupivacaine combined with epinephrine, liposomal bupivacaine had a significantly lower incidence of seizures and ventricular tachycardia [29]. Therefore, multilayer liposomal bupivacaine may reduce neurological and cardiac toxicity during intravascular infusion. Clinical trials have investigated regular preparations of local anesthetics combined with epinephrine and local anesthetic liposomes separately injected into the oral cavity. Their pharmacokinetic results showed that local anesthetic liposomes allowed similar systemic absorption as that of the regular preparation and reduced injection discomfort [34]. Accordingly, local anesthetic liposomes can be a safer alternative for local anesthesia [35] with prolonged anesthetic effects [33].

In summary, combining nanotechnology with local anesthesia could allow accurate administration of the local anesthetic drug, slow drug release, prolonged anesthetic effects, and reduced toxicity. Specifically, it can improve the safety and longevity of local anesthesia in cardiac terms, as well as overcome the aforementioned limitations of traditional local anesthesia. Finally, it could allow more precise control of the local anesthesia range.

The primary dosage forms of commercially available local surface anesthetics include gels, ointments, and creams [59]. However, their poor anesthetic penetration and susceptibility to allergies and other adverse effects can adversely affect the patient experience. Liu et al. [57] performed comparative in vitro and in vivo mouse experiments using bupivacaine (TTC) poly (L-lactide) nanoparticles (PLA NPs), solid lipid nanoparticles (SLNs), and NLCs. TTC NLCs showed the most optimal in vivo efficiency with respect to improved skin penetration, analgesia duration, and pain control intensity. Contrastingly, TTC PLA NPs and TTC SLNs showed the best serum stability and in vitro penetration efficiency, respectively. Nonetheless, TTC NLCs showed the most promise for long-term anesthetic analgesia. Another study on liposomal and non-liposomal lidocaine (LDC) preparations administered to the palmar side of the forearm of healthy volunteers found that liposomal LDC preparations allowed a longer anesthesia duration [26]. Further research is warranted to inform the development and clinical application of liposomal local anesthetics, which can provide high permeability, stability, low treatment cost, and a shortened hospital stay with a prolonged duration of local surface anesthesia [19].

Additionally, there have been recent studies on new solutions in hybrid nanomembranes for reducing the fear of needling and pain in dental patients. Lígia et al. [24] reported that the most effective formulation was based on pectin (2%) as a biopolymer and nanostructured lipid carrier, followed by 5% LDC-PLC encapsulated in a lipid matrix. Moreover, nanohybrid membranes for LDC-PLC delivery allowed for more prolonged anesthetic effects, improved drug penetration, safer anesthetic conditions, flexibility, and good mucosal adhesion. This indicates the potential clinical application of hybrid nanomembranes in dental anesthesia, which may allow a more comfortable anesthesia experience for patients with a fear of needling and those with low pain thresholds. Moreover, they could allow precise, unique, and individualized medical treatment. Notably, a clinical study reported that liposomal PRC encapsulation did not have significantly better anesthetic effects than a regular solution for maxillary infiltration anesthesia [31]. However, further research is warranted.

Ophthalmic anesthesia often requires high-dose and frequent topical administration and provides high drug bioavailability. Nanoparticle systems can improve drug delivery to the anterior and posterior eye segments. Levobupivacaine, ropivacaine, and bupivacaine nanoliposomes are characterized by high biocompatibility, biodegradability, and low toxicity [19]. Moreover, tenofovir nanoparticles have increased membrane permeability [60], and the microemulsion system of docetaxel has increased cellular uptake potential [61]. Furthermore, nano-emulsions can dissolve hydrophobic and hydrophilic drugs with improved chemical stability, shelf life, and therapeutic efficacy. Taken together, liposomes, solid nanoparticles, polymeric hydrogels, and nano-emulsions can confer several advantages in ophthalmic anesthesia. Moreover, nanotechnology has been incorporated into contact lenses [62,63,64] and has also been added to retinal neuroprotection therapies [65].

#### 2.1.3. Nanotechnology in Perioperative Pain Management

Perioperative pain mainly results from tissue damage caused by surgical incision, separation, cautery, or direct nerve injury. The release of local inflammatory mediators enhances sensitivity to stimuli in the area surrounding the injury, which is known as nociceptive hypersensitivity. Alternatively, patients may incorrectly perceive pain in response to non-injurious stimuli [66,67,68,69]. Traditionally, the management of acute perioperative pain has solely relied on opioids targeting nociceptive-related centers. However, this has been limited by inappropriate use, tolerance, abuse, and opioid addiction [70]. The ongoing opioid crisis has highlighted the need for non-steroidal anti-inflammatory drugs (NSAIDs), and nanoformulations of NSAIDs with higher efficacy or lower toxicity remain an important and innovative breakthrough point in analgesic and anti-inflammatory treatment [71]. New drugs and techniques have been added to analgesic protocols, which has resulted in multimodal analgesia and improved treatment. It is worth mentioning that despite the therapeutic importance of these drugs, they may present a paradoxical effect of generating allodynia phenomena due to their constant use or overdosage [72].

In addition to nanoformulations of non-steroidal anti-inflammatory drugs being used in pain management, other drugs, such as MgO, MnO_2_, Fe_3_O_4_, gabapentin, amantadine, and cannabinoids, also have a place in this field [73]. For example, in the hot plate test conducted on Wistar rats [74], it was found that zinc oxide nanoparticles exhibited a higher analgesic effect compared to bulk zinc oxide. Jahangiri et al. [75] demonstrated through testicular experiments that MgO nanoparticles have a stronger analgesic effect compared to conventional MgO. MnO_2_ can influence the dopaminergic system, regulating analgesia and pain perception. The effect of MnO_2_ on pain threshold in rats was observed using the tail immersion method. The results showed that both nano-sized MnO_2_ and micro-sized MnO_2_ exhibited good corrosion resistance [75]. To date, there have been few studies reporting the analgesic effects of Fe_3_O_4_ nanoparticles. Wu et al. [76] established an inflammatory pain model in CD1 mice by injecting complete Freund’s adjuvant. The results showed that local administration of Fe_3_O_4_ nanoparticles could inhibit macrophage activity, inflammatory cells, and pro-inflammatory markers, leading to analgesic effects. Furthermore, it significantly reduced the production of reactive oxygen species (ROS) in the injured paws. In addition, in traditional Chinese medicine, curcumin nanoparticles in various forms have been developed to prolong the release and therapeutic effects of curcumin, providing us with a strategy to combine non-steroidal drugs with nanotechnology to extend analgesic effects [77].

However, unpleasant sensory and emotional pain experiences have persisted, which adversely affect the disease prognosis, physical function, daily activities, and psychological and mental status [78]. Accordingly, accurate and effective pain management using multimodal analgesia is difficult [79,80]. Further integration of nanotechnology with clinical medicine may yield novel strategies for pain management [81]. The use of drug-loaded nanoparticles holds promise for the treatment of both acute and chronic pain conditions [82]. Dong et al. [83] reported that in a rabbit model of knee osteoarthritis, intra-articular injection of selexipag liposomes encapsulated in hyaluronic acid gel showed superior efficacy in pain control and cartilage protection compared to any single drug alone. An observational study reported that the intensity of postoperative pain and unacceptable pain were positively correlated with the incidence of postoperative complications, even on the first postoperative day [84]. In 1014 patients, 55% experienced moderate-to-severe pain on the first postoperative day. The overall complication rate was 34% [84]. This suggests that efficient pain management could improve the postoperative prognosis. A randomized, double-blind, controlled study reported that among patients aged 18–86 undergoing hemorrhoidectomy, compared with a placebo, bupivacaine extended-release liposome injection allowed significantly lower pain scores, fewer opioid requirements, a later onset of first opioid use, and higher patient satisfaction up to 72 h after hemorrhoidectomy [85]. Furthermore, a new liposomal bupivacaine formulation has been shown to yield ultra-long-lasting analgesia with prolonged dose-dependent analgesic duration [86]. These findings suggest that applying nanotechnology to pain management could allow a prolonged analgesic effect, increase the effectiveness of the analgesics, and limit the use of opioids. Validation of these findings through clinical trials could inform the clinical application of nanotechnology in pain management and allow precision pain therapy.

Contrastingly, a study conducted reported that extended-release liposomal bupivacaine did not decrease opioid use or cumulative postoperative vaginal pain on days 1 and 3 after posterior vaginal wall surgery [87]. Accordingly, the effectiveness of nanoformulations for analgesic treatment remains unclear, and further studies are warranted.

Manual anesthetic administration by the anesthesiologist yields postoperative epidural analgesia toward the end of the procedure. Alternatively, an epidural catheter connected to an epidural infusion pump containing a low concentration of local anesthetic/opioid solution can be used to prevent the patient from experiencing increased postoperative pain. For example, this technique is applied in the management of postoperative pain in general or visceral surgery, vascular and thoracic surgery, gynecology, urology, and orthopedic surgery [88]. However, motor and sympathetic blocks may allow for more extensive epidural analgesia. Further, adverse effects such as Horner’s syndrome, apnea, and loss of consciousness may occur [89]. A single-center study involving over 25,000 cases of postoperative epidural analgesia demonstrated a slightly higher failure rate for thoracic epidural analgesia compared to lumbar epidural analgesia (32% vs. 27%). Failure was defined as the need for reinsertion of the epidural catheter or the addition of other major analgesic modalities, such as patient-controlled intravenous analgesia [90]. Possible reasons for the failure could include a lack of training or insufficient equipment. A study conducted grouped 26 patients to postoperatively receive 0.5% bupivacaine and 1:200,000 epinephrine alone or liposomal 0.5% bupivacaine epidural analgesia. The liposomal bupivacaine formulation increased the duration of analgesia without motor blockade or adverse side effects [91]. Lafont et al. [92] reported that a liposomal formulation for epidural analgesia produced a much higher analgesic effect than a simple solution in a patient with cancer-related pain. Moreover, it was effective for up to 11 h without motor blockade or hemodynamic instability. In addition, a 6-week bupivacaine liposomal brachial plexus infiltration treatment was found to achieve complete pain resolution in a 32-year-old female patient with chronic arm pain [93].

Multimodal analgesia is defined as the utilization of two or more analgesics from different pharmacological classes, targeting multiple sites in the neurobiology of pain. The aim is to inhibit pain perception and improve pain management so it can act at different levels of the pain recognition pathway [94,95]. Peijun et al. [96] reported that the SLN systems of LDC and PRC had better ex vivo skin penetration than NLC. Contrastingly, the NLC system yielded stronger in vivo anesthetic/analgesia than the SLN system. Notably, SLN and NLC co-loaded with LDC and PRC allow enhanced skin penetration and analgesic effects. Similarly, a dual drug delivery system showed more efficiency than a single delivery system. The ex vivo skin permeation efficiency of LID and/or PRI-loaded SLNs and NLCs was much higher than that of free drug solutions (*p* < 0.05). The in vivo TF latency test is the most frequently used method to assess anesthesia depth. LID- and/or PRI-loaded SLNs and NLCs showed a longer-lasting effect than the free drug groups. The results indicated that the drug-loaded nanocarriers revealed a more interesting anesthetic effect in the first few minutes and displayed sustained anesthetic activity compared with free drugs. The more remarkable anesthetic effect of the NLC systems than the SLNs illustrates that the impressive anesthetic effect of the NLC systems could bring about better therapeutic effects than the SLNs [96]. Transcriptional trans-activator (TAT)-modified nanostructured lipid vectors (TAT-NLC) have been modified to co-deliver meloxicam (MLX) and ropivacaine (RVC) in a TAT-NLC-RVC/MLX system. Animal studies have demonstrated that this TAT-NLC-RVC/MLX system showed better systemic stability, lower cytotoxicity, more prolonged analgesic effects, and higher drug penetration efficiency in vivo and in vitro [97,98]. Furthermore, MLX exerts anti-inflammatory effects, which may contribute to reducing pain in rats [99]. Accordingly, the TAT-NLC-RVC/MLX system allows improved pain management with reduced inflammation in the injured area [97]. Taken together, dual drug delivery can achieve a synergistic analgesic effect and apply the advantages of each drug. This strategy could inform the effective management of postoperative and general pain.

## 3. Genomics and Anesthesia

Genomics, which is a fundamental component of precision medicine, allows the elucidation of disease onset and progression at the molecular and cellular levels. With advances in sequencing technology and molecular biology, genomic-related science influences all aspects of medicine. Genomics may improve the effectiveness of treatment, promote diagnostic certainty, and allow the prediction of disease susceptibility to achieve tailored diagnosis and treatment [100,101].

### 3.1. Pharmacogenomics

Pharmacogenomics is the branch of pharmacology that uses genomic information to decipher individualized differences in drug action as a way to study the relationship between human genomic information and drug response (sensitivity, metabolism, and adverse effects). Anesthetics and their adjuvants are essential for surgical procedures. However, drug responses under routine clinical regimens have shown significant differences across individuals, with studies showing that the same standard drug dose may result in plasma drug concentrations varying by up to 1000 to 10,000 units across individuals [102]. Genetic variations result in the variability of responses by affecting drug metabolism, transport, and targets [103]. Accordingly, pharmacogenomics allows the investigation of the mechanisms through which genetic variations contribute to such differences. Pharmacogenomics may inform personalized therapy for improved efficacy and safety [104].

This section focuses on cytochrome P450 (CYP450), which is responsible for almost 80% of the first-phase metabolism of currently used drugs (Table 2) [105]. CYP450 is mainly distributed in the endoplasmic reticulum of hepatocytes and enterocytes [106], with three subfamilies, CYP1, CYP2, and CYP3, playing an important role [107].

#### 3.1.1. CYP1

CYP1A2 enzymes metabolize approximately 10% of clinically used drugs [133], including phenacetin, caffeine, clozapine, tacrine, propranolol, and mexiletine, as well as some endogenous compounds such as melatonin and estradiol. In a study on caffeine metabolism, Nut et al. reported that 163C > A (rs762551), 3860G > A (rs2069514), 2467delT (rs356941), and 3113A > G (rs2069521) are related genetic variants that contribute to differences in the degree of CYP1A2 activity [134]. The pathogenic cytochrome P450 oxidoreductase (POR) mutations A287P and R457H reduce the catalytic effect of CYP1A2, while the Q153R mutation increases CYP1A2 activity to 144% of the normal level [135].

#### 3.1.2. CYP2

CYP2 is the largest family of CYP450 enzymes, including CYP2A, CYP2B, CYP2C, CYP2D, and CYP2E subfamilies [136], of which CYP2C9, CYP2C19, and CYP2D6 are the enzyme families mainly associated with the metabolism of clinical anesthetics and their adjuvants. They have the highest genetic polymorphism in the CYP2 family [137] and are responsible for the metabolism of 40% of drugs [136], such as anticoagulants [138,139], benzodiazepines [140], opioids [141], antiemetics [142] and β-blockers [143].

##### CYP2C9

Multiple genetic variants of CYP2C9 and VKORC1 are related to the anticoagulant effect of warfarin [144], which could explain the wide variations in the warfarin dose requirements across individuals. CYP2C9 is primarily involved in warfarin metabolism. Further, the S-isomer of warfarin is almost exclusively metabolized by CYP2C9 to an inactive 7-hydroxy metabolite [139]. The CYP2C9*2 and CYP2C9*3 alleles can significantly enhance warfarin sensitivity [139]. Accordingly, gene carriers require reduced doses to prevent an increased bleeding risk, as demonstrated by a study on the clinical use of warfarin anticoagulants in Chinese patients with coronary artery disease [145]. In 2007, the FDA included pharmacogenetic data in the product labeling of warfarin and provided dosing recommendations based on known genotypes.

##### CYP2C19

Unlike warfarin, clopidogrel’s metabolism is influenced by CYP2C19 gene polymorphisms. Additionally, CYP2C19*2 and CYP2C19*3 can lead to a significant decrease or even loss of enzyme activity [146]. Jessica et al. [147] reported that among carriers of even one reduced-function CYP2C19 allele, percutaneous coronary intervention with clopidogrel was related to a significantly increased risk of major adverse cardiovascular events, especially stent thrombosis [148]. Moreover, CYP2C19 is involved in the metabolism of benzodiazepines, with the most commonly used in clinical practice being midazolam and diazepam. The FDA-approved drug label for diazepam gel has the following statement: “The significant interindividual variability in diazepam clearance reported in the literature may be attributable to variability in CYP2C19.” Shinichi et al. [140] genotyped individuals as extensive metabolizers (no variant, *1/*1) and poor metabolizers (2 variants, *2/*2, *2/*3, or *3/*3) and observed that the difference in the area under the plasma concentration curve (AUC) over 24 h could even reach 1000 ng/mL [140]. Recent studies have demonstrated that poor CYP2C19 metabolizers have a two-fold higher AUC compared with normal metabolizers. It is clinically recommended that a 25–50% dose reduction be applied to CYP2C19 poor metabolizers to avoid adverse drug reactions or tolerance [148].

##### CYP2D6

CYP2D6 significantly contributes to the metabolism of 20–25% of clinically used drugs [149], including opioids (codeine, tramadol, oxycodone), antiemetics (ondansetron, toltesetron), and β-blockers (metoprolol, propranolol, timolol). Based on the number of CYP2D6 alleles, patients can be classified as poor metabolizers (PMs), intermediate metabolizers (IMs), ultra-rapid metabolizers (UMs), and extensive metabolizers (EMs).

Opioids

CYP2D6 metabolizes codeine to morphine, which exerts analgesic effects. Enhanced drug metabolic responses and increased morphine formation in UMs can lead to an increased risk of toxicity [141]. Furthermore, there have been reported cases of deaths among breastfed newborns of UM mothers who were exposed to toxic morphine levels after codeine administration. Comparatively, codeine exerts insufficient analgesic effects in PMs, which is consistent with the findings by Sondra et al. regarding post-cesarean pain [149]. The Clinical Pharmacogenetics Implementation Consortium (CPIC) guidelines indicate that codeine treatment should be avoided in UMs and PMs; instead, alternative analgesics such as morphine should be considered [150]. Similar to codeine, tramadol is metabolized by CYP2D6 to Odesmethyl tramadol (M1), which exerts its analgesic effects. Physiological pharmacokinetic modeling studies of tramadol have demonstrated that the area under the concentration–time curve (AUCinftDlast) is 70% lower in PMs and 15% higher in UMs than in EMs [151].

Antiemetic

5-hydroxytryptamine receptor antagonists such as ondansetron and tropisetron are often used as antiemetics, with CYP2D6 playing a minor role in ondansetron hydroxylation. Contrastingly, up to 91% of tropisetron metabolism is performed by CYP2D6 [133]. There is clinical evidence that CYP2D6 UMs may experience a decreased antiemetic effect of tropisetron. Therefore, alternative drugs that are not metabolized by CYP2D6 (e.g., granisetron) are recommended for UMs by the CPIC [142].

Beta-blockers

Many beta-blockers are CYP2D6-metabolizing substrates, with metoprolol being the most dependent (≈70% of the drug metabolism). CYP2D6 PMs showed more pronounced reductions in diastolic blood pressure, systolic blood pressure, and heart rate following treatment with metoprolol, as well as an increased risk of bradycardia [143]. Moreover, a prospective observational study showed that CYP2D6 PMs had six times higher plasma trough concentrations of metoprolol than EMs (who represented the majority of the population) [152]. The Dutch Pharmacogenetics Working Group guidelines recommend a 70–75% reduction in the metoprolol dosage in PMs [153]. Further, the CYP2D6 genotype can be used to determine the maintenance dose for patients taking β-blockers. Accordingly, Choong-Min et al. developed a pharmacokinetic prediction model based on the CYP2D6 genotype, which allowed genetically personalized drug therapy using metoprolol. Timolol therapy has been shown to considerably decrease the heart rate in PMs (especially during exercise) [154]. Contrastingly, although propranolol is metabolized via CYP2D6, its plasma concentration does not appear to be affected by the genotype.

#### 3.1.3. CYP3

There are four CYP3A isoforms, including CYP3A3, CYP3A4, CYP3A5, and CYP3A43. CYP3A4 is the most critical form of P450 expressed in the normal adult liver and metabolizes up to 50% of all clinically used drugs. Its important narcotic-related substrates include oxycodone, ketamine, midazolam, and other drugs. CYP3A metabolizes oxycodone into fewer active metabolites. Although CYP3A inhibitors significantly alter the in vivo exposure of oxycodone metabolites [155], variants in the CYP3A4 and CYP3A5 loci resulting in altered enzyme activity are rare. Further, there are no reports of altered oxycodone responses in the presence of these variant alleles. Regarding ketamine, CYP3A is only involved in its high-level metabolism (not clinically relevant plasma levels) [156]. Although genetic factors crucially contribute to individual differences in CYP3A activity for oxycodone and ketamine, the influence of the genetic phenotype is usually ignored. Midazolam is a common sedative drug whose drug metabolism and pharmacological effects are altered by genetic changes in CYP enzymes. From a genetic perspective, compared with CYP3A4 or CYP3A5 [157,158], POR*28 has a much greater impact on CYP3A activity. Among CYP3A5-expressing patients, the ratio of midazolam metabolism is 45% lower in POR*28 carriers than in POR*1 carriers [159]. Interestingly, midazolam is used as a metabolic substrate for CYP3A in clinical studies; additionally, changes in CYP3A activity are often clinically measured based on midazolam metabolism [160].

Nonetheless, drug metabolism is influenced by several other enzymes, and this section only discusses the main metabolic enzymes of related drugs. Genomics can allow the elucidation of individual differences in patient drug metabolism, which can consequently inform precision medicine in clinical anesthesia.

### 3.2. Disease Genomics

Precision medicine in anesthesia warrants the use of different perioperative anesthesia strategies according to the characteristics of patients with different diseases. Muscular dystrophy is a complex and diverse genetic muscle disease caused by mutations in >40 genes [161]. It is often characterized by progressive muscle weakness involving the heart and respiratory system. It is important to closely consider ankylosing muscular dystrophy caused by DMPK and CNBP gene mutations in order to prevent reflux aspiration since its patients mainly present gastrointestinal motility dysfunction [162]. Further, opioid use should be cautiously considered given their increased drug sensitivity, which may lead to the risk of complications, as indicated by The European Neuromuscular Center consensus statement on anesthesia for patients with neuromuscular diseases [163]. Contrastingly, monitoring respiratory insufficiency is crucial for patients with congenital muscular dystrophy. Moreover, among these patients, carriers of LMNA, COL6A, LAMA2, or SELENON variants should be monitored for difficult airways resulting from difficulties in cervical spine movements [164]. Myotonic dystrophy, Emory-Dreyfus myotonic dystrophy, and limb–girdle myotonic dystrophy type 1B are associated with potentially fatal arrhythmias. Monitoring and prompt intervention with a defibrillator could significantly improve perioperative patient safety. In addition, regarding pharmacogenomic-related diseases, clinical studies have shown that CACNA1S, RYR1, or STAC3 variant carriers have an increased susceptibility to malignant hyperthermia [165]. Accordingly, volatile anesthetics and depolarizing muscle relaxants should be avoided or used cautiously. Similarly, patients with butyrylcholinesterase deficiency who carry BCHE gene variants (mainly A, K, F1, F2, and S15) may exhibit increased sensitivity to muscle relaxants and prolonged postoperative paralysis. Therefore, related muscle relaxants should be used cautiously to avoid serious complications. It is worth noting that the use of antibiotics may also lead to the development of myasthenia gravis (MG) or MG-like symptoms [166]. Antibiotics that have been identified with similar complications are macrolide antibiotics [167], quinolone antibiotics [168], aminoglycoside antibiotics [169,170], penicillin [171], lincosamide antibiotics [172], and polymyxin B [173]. Most of their mechanisms of action are impairment of pre- and postsynaptic neuromuscular signaling, inhibition of neurotransmitter release, competitive blockade of nAChR channel currents, etc. [174]. There are no clear genomic studies related to antibiotic-induced MG, but in a genome-wide association study of myasthenia gravis by Alan E Renton et al. [175], three genes, CTLA4 (rs231770), HLA-DQA1 (rs9271871), and TNFRSF11A, were found to be closely associated with myasthenia gravis [175].

### 3.3. Decision Modeling

Extensive application of genomics and data analysis has facilitated steps toward precision medicine. Tien et al. developed an ImPreSS project model for preemptive gene sequencing to explore the feasibility and potential for individualized perioperative patient care by anesthesia or critical care physicians [176]. In addition, in pediatric critical care, Mestek-Boukhibar et al. developed a rapid reporting system using whole genome sequencing to screen for rare diseases, which directly informed the clinical management of 3 out of 24 critically ill children [177]. The development of assisted decision-making systems based on genomics could inform the clinical application of genomic medicine.

## 4. Research and Application of 3D Printing in the Anesthesia Field

### 4.1. Three-Dimensional Printing Technology

Three-dimensional printing is a type of additive manufacturing technology. It involves the digital structuring of models using bondable materials, including metals, plastics, or polymers. Here, objects are constructed by printing layer by layer in a stacked manner [178]. Three-dimensional printing allows for free design and mass customization of complex structures using a computer, as well as rapid prototyping with digital material printers. Further, it facilitates waste minimization [179]. Three-dimensional printing has evolved from its initial industrial applications to revolutionary applications in aerospace, architecture, and protective structures [179]. Moreover, 3D printing is being gradually applied in biomedical research and clinical medicine, given its aforementioned advantages [180].

Three-dimensional printing can allow the simplification of specific, complex therapeutic problems in clinical settings. In 2002, Jill et al. [181] used 3D-printed cartilage composite scaffolds to repair articular cartilage. Furthermore, the tensile strength of the formed cartilage was similar to that of fresh cancellous human bone formed through in vitro culture. Other studies have placed 3D-printed stentless artificial tracheas in mice [182] as well as high-precision, flexible, and biodegradable tracheal stents in rabbits [183]. Taken together, 3D printing may allow rapid manufacturing of customized medical devices with desirable mechanical and biological properties when applied in vivo.

### 4.2. Three-Dimensional Printing and Anesthesia

Anesthesiologists often require strong clinical thinking and clinical experience to handle distinctive emergency cases with different conditions. Ultrasonography can allow accurate visualization of specific nerves and tissues when administering regional anesthesia for improved safety and effectiveness [13]. Similarly, 3D printing can improve the precision and personalization of anesthesia. Specifically, it can directly replicate anatomical structures from medical images into functional anatomical simulators. Moreover, it can combine realistic tactile feedback, repeatability, and the potential for patient-specific pathology modifications [184]. Finally, it provides training opportunities for anesthesiologists to improve their proficiency in anesthesia operations. The distinct advantages of 3D printing, including adjustable combination patterns, rapid prototype replication, and low cost, facilitate their use in clinical anesthesia operations.

#### 4.2.1. Three-Dimensional Printing and Anatomical Models

Tracheal intubation and airway management are crucial for ensuring the perioperative stability of patients’ vital signs. However, airway control operations involve certain risks. An observational study on tracheal intubation in critically ill patients at 197 sites in 29 countries reported the frequent occurrence of adverse peri-intubation events, including cardiovascular instability, severe hypoxemia (9.3%), and cardiac arrest (3.1%) [185]. Three-dimensional printing could allow the simulation of the tracheal anatomy of critically ill or unique patients and inform tailored treatment protocols.

Furthermore, 3D printing allows indirect visualization of the procedure, which enhances operational proficiency and airway control success and thus reduces deaths due to procedure-related complications. However, the airway anatomy and specific medical conditions differ across populations and individuals. Accordingly, the risks related to airway control operations differ among patients. Three-dimensional printing is customized for airway planning in different populations, including infants [186], pediatric patients [184,187,188], and adult patients, even in specific cases involving congenital or acquired craniofacial anomalies [15], difficult airway intubation [189], cricothyroid puncture [190], thoracic puncture [180], thoracic epidural analgesia [191], or control of combined lumbar and epidural anesthesia planes.

A previous study developed an in-house modified manikin for extracorporeal cardiopulmonary resuscitation. The manikin comprised a modular, internally designed extracorporeal membrane oxygenation cannula and vascular structure. Further, it features an ultrasound view, intubation, and functional resuscitation components combined with commercially available airway and cardiopulmonary resuscitation components. This manikin could improve simulation exercise proficiency for first responders, paramedics, and emergency and critical care physicians [192].

A previous study reported the use of 3D printing and virtual reality to develop a personalized airway plan for a 7.5-year-old child [193], which allowed accurate elucidation of the child’s airway structure and improved the safety and success of the procedure. Three-dimensional printing allows visualization of the oral nerve alignment and anesthetization of specific oral regions, which could reduce patient discomfort, the risk of nerve injury, mandibular anesthesia failure, and the total anesthetic dose [194].

#### 4.2.2. Three-Dimensional Printing and Anesthesia Equipment

Anesthesia equipment plays a pivotal role as an auxiliary tool. Three-dimensional-printed anesthesia equipment and tools are more economical and convenient. A previous study described the use of 3D printing technology to manufacture a new syringe holder, which conferred numerous safety advantages [195]. Additionally, 3D printing has been used to develop novel, sustainable, human-powered, low-cost thermal laryngoscopes [196,197]. Taken together, the speed, ease, low cost, and customizability of 3D printing technology can allow the planning of regional anesthesia for specific areas.

## 5. Future, Limitations, and Outlook of Precision Anesthesia

Technological advances and the advent of the era of big data are allowing progress toward precision medicine. Moreover, genomics with highly sophisticated technology can inform accurate and appropriate clinical decisions by anesthesiologists. Applying nanotechnology in clinical anesthesia improves efficiency, safety, and comfort. Additionally, the economical, fast, convenient, and customizable nature of 3D printing technology confers numerous advantages in clinical anesthesia. However, most of these advanced technologies in the field of anesthesia are still at the stage of animal or clinical studies. Accordingly, further research is warranted to facilitate their clinical application. In addition, the evidence described in this article is limited to anesthesia. It does not extensively cover some anesthesia-related areas, including general anesthesia as well as platelet and red blood cell transfusion [18]. However, there are currently ongoing clinical studies on more advanced technologies. Integrated research encompassing all medical specialties could facilitate the scientific elucidation of the nature of human functions and diseases. Moreover, it could allow optimization of the prevention and treatment of diseases across different conditions, populations, and individuals. Finally, this could maximize individual and societal health benefits through efficient, safe, and cost-effective healthcare services, as well as help establish a new paradigm of healthcare services.

## 6. Conclusions

In conclusion, the integration of nanotechnology with clinical anesthesia has the potential to advance precision medicine toward more personalized and precise approaches in inhalation anesthesia, local anesthesia, and pain management. Additionally, the combination of genomics and 3D printing in the field of clinical anesthesia can contribute to better fulfilling the medical needs of individual patients with greater precision.

## Figures and Tables

**Figure 1 pharmaceutics-15-02289-f001:**
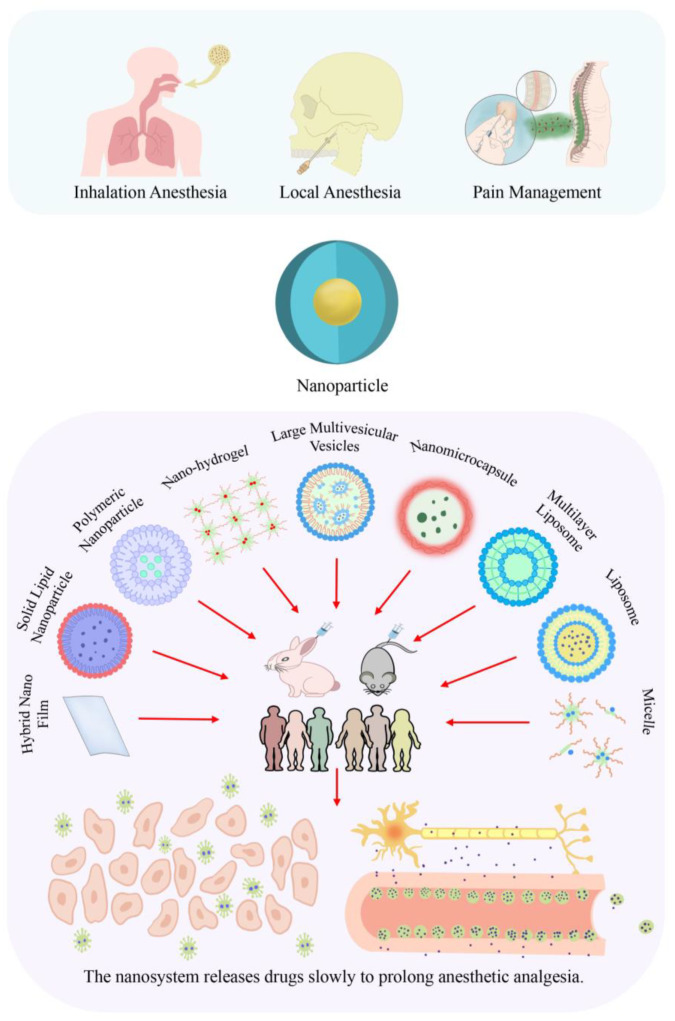
Application and classification of nanoformulations and the slow-release effect in drug delivery. Different nanocarriers exhibit varied drug release effects and application methods. For instance, Hybrid Nano Film can be used to prolong the surface anesthesia effect on the skin [24], and micellar systems can extend the duration of local intravenous anesthesia [25]. At the same time, Liposomes can be employed for both surface anesthesia on the skin [26] and local intravenous anesthesia [25,27,28,29,30,31,32,33,34,35].

**Figure 2 pharmaceutics-15-02289-f002:**
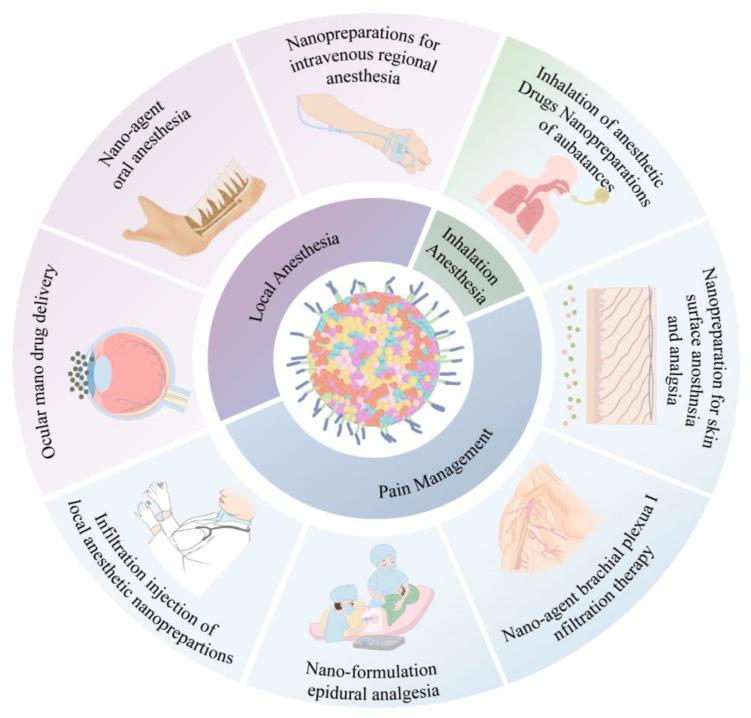
Application of nanoformulations in anesthesia.

**Table 2 pharmaceutics-15-02289-t002:** Pharmacogenomics of commonly used anesthetic drugs.

Drug Classification	Representative Drugs	CYP450 Metabolizing Enzymes [108,109,110,111,112,113]		Other Special Genes
		Related Genes	Related Variant Subtypes	
Local anesthetics	Lidocaine Ropivacaine	CYP1A2 CYP3A4 CYP3A5	CYP1A2 ^a^: Thr83Met, lu168Gln, Phe186Leu, Ser212Cys, Gly299Ala, hr438Ile [114]	SCN5A ^b^ [115] MC1R ^c^ [116]
Benzodiazepines	Midazolam Diazepam	CYP2C19 CYP3A4 CYP3A5	CYP2C19 *2/*3/*17 CYP3A4 *1B CYP3A5 *1/*3/*6/*7	-
Inhalation of narcotics	Halothane Sevoflurane Isoflurane	CYP2E1	CYP2E1 *1A/*5B/*6/*7B	RYR1 ^d^ [117] MC1R [118]
Opioid analgesics	Fentanyl	CYP2D6	CYP2D6 1/*2/*3/*4/*5/*6/*10/*17/*35/*41 [119] CYP3A4 *1/*1G CYP3A5 *1/*3 [120] CYP2B6 *6	COMTb ^e^ [121] UGTb ^f^ [122] ABCB1b ^g^ OPRM1b ^h^ [123] OPRK1b ^i^ MDR1 [124]
	Codeine	CYP2B6		
	Morphine	CYP3A4		
	Tramadol	CYP3A5 CYP1C2 CYP1D2 CYP2B11 CYP2C41 CYP2D2 CYP2D15 [125,126]		
Intravenous anesthetics	Propofol	CYP2B6 CYP2C9	CYP2B6 *4/*6 CYP2C9 *2	UGT1A9 [111] GABA [127]
Non-steroidal anti-inflammatory drugs	Aspirin Celecoxib	CYP2C8 CYP2C9	CYP2C8 *1/*2/*3 CYP2C9 *1/*2/*3	PTGS1 PTGS2 ^j^ [128]
Neuromuscular blocking drugs	Succinylcholine Vecuronium-Bromide Rocuronium	CYP3A4 CYP2C19	-	BCHE ^k^ [129] SLCO1B1 ABCB1 [115] RYR1 nAChR ^l^ [130]
Anticoagulants	Warfarin Clopidogrel	CYP2C9 CYP2C19	CYP2C9 *1/*2/*3 CYP2C19 *1/*2/*3/*9/*12/*14/*17 [130]	VKORC1 ^m^ [131]
Antiemetic	Tropisetron Granisetron	CYP2D6 CYP3A4	CYP2D6 *1/*2/*3/*4/*5/*6/*9/*41	5-HT3B [132] ABCB1 SLC22A1 ^n^

^a^: The genetic polymorphism of CYP1A2 is mainly expressed within the gene; ^b^: encodes sodium channels; ^c^: encodes melanocortin 1 receptor; ^d^: encodes the ryanodine receptor; ^e^: encodes catechol-O-methyltransferase; ^f^: encodes uridine 5-diphosphate glucuronosyltransferase; ^g^: encodes adenosine triphosphate-binding cassette transporter; ^h^: encodes mu-opioid receptors; ^i^: encodes kappa receptors; ^j^: encodes cyclooxygenase 1 and cyclooxygenase 2; ^k^: encodes butyrylcholinesterase; ^l^: encodes nicotinic acetylcholine receptors; ^m^: encodes a vitamin K cyclic oxidoreductase complex; ^n^: encodes a transporter for related drugs.

## Data Availability

No new data were created or analyzed in this study. Data sharing is not applicable to this article.

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
