# Peer review of "Application of Advanced Technologies—Nanotechnology, Genomics Technology, and 3D Printing Technology—In Precision Anesthesia: A Comprehensive Narrative Review"

_pharmaceutics, 2023, doi:10.3390/pharmaceutics15092289_

Round 1

Reviewer 1 Report (Previous Reviewer 4)

I thank the authors for considering my comments in the first revision of their manuscript. It seems to me that the article has improved substantially, so I am convinced that if this review material is published, readers will have an important contribution to favor the development of Clinical anesthesia, particularly in the field of the Application of Advanced Technologies; Nanotechnology, Genomics Technology, and 3D Printing Technology in Precision Anesthesia.

However, in the new manuscript, I have found some aspects that require the attention of the authors before publishing the article.

L59: please delete the reference. This same citation is indicated in L63.

L70-72: Narrative reviews also have a methodology for searching and collecting information, so it is necessary to indicate the sites or sources used to search for information, keywords used in the search, inclusion and exclusion criteria taken into account, or even how the authors were able to classify the articles consulted according to their content. I suggest to the authors that this information be added concretely in a paragraph.

In Table 2 with respect to other genes and variants related to tramadol; references 121 and 122 emphasize drugs such as remifentanil and morphine. Moreover, references 123 and 124 are not indicated in the table. Please check this aspect.

Best regards

Author Response

1.The reviewer’s comment: L59: please delete the reference. This same citation is indicated in L63. Response: Apologies for the repetition. I have removed the reference from line 59 to avoid duplication. 2. The reviewer’s comment: L70-72: Narrative reviews also have a methodology for searching and collecting information, so it is necessary to indicate the sites or sources used to search for information, keywords used in the search, inclusion and exclusion criteria taken into account, or even how the authors were able to classify the articles consulted according to their content. I suggest to the authors that this information be added concretely in a paragraph. Response: Thank you for your kind suggestion. Including information about the methodology used in conducting the narrative review, such as search strategies, inclusion/exclusion criteria, and article classification, can enhance the transparency and rigor of the study. I have made modifications to lines 69 to 73 of the article. 3. The reviewer’s comment: In Table 2 with respect to other genes and variants related to tramadol; references 121 and 122 emphasize drugs such as remifentanil and morphine. Moreover, references 123 and 124 are not indicated in the table. Please check this aspect. Response: I apologize for the confusion. I have made the modifications in the article. Thank you once again for your suggestions.

Reviewer 2 Report (Previous Reviewer 3)

I am grateful to the authors for adopting and making the suggested changes before re-submitting this version of the manuscript. I also commend the authors for improving the content and structure of the present review which addresses a very interesting and innovative application of nanotechnology in medicine. I left some remaining style-related comments below.

Lines 25-26: Consider removing this sentence since its main idea is repeated in the subsequent lines but with different words.

Response:

Line 83: Please, revise the Instructions for authors to amend the Figure citation.

Response:

Line 123: Replace “However” with “Therefore, …”

Response:

Line 171: Replace “However” with “In contrast,”

Response:

Line 260: Please, remove “Additionally” and start the sentence from “The release of local inflammatory…”

Response:

Line 275: From “MnO2” and “Fe3O4”, put the numbers in subscripts. Apply this change in the subsequent lines.

Response:

Line 541: Amend citation style (e.g., Renton et al. [173]).

Response:

Author Response

1. The reviewer’s comment: Lines 25-26: Consider removing this sentence since its main idea is repeated in the subsequent lines but with different words. Response: Thank you for your suggestion. The sentence in lines 25-26 is repetitive and restated in subsequent lines with different wording, I have removed it to avoid redundancy. 2. The reviewer’s comment: Line 83: Please, revise the Instructions for authors to amend the Figure citation. Response: Thank you for your kind suggestion. I have provided additional clarification in lines 88 to 89 of the article. Thank you again for your helpful suggestion. 3. The reviewer’s comment: Line 123: Replace “However” with “Therefore, …” Response: Thank you for your kind suggestion. Replacing "However" with "Therefore" in line 123 can help strengthen the logical flow of the sentence. I have made modifications to line 129 of the article. 4. The reviewer’s comment: Line 171: Replace “However” with “In contrast,” Response: Thank you for your kind suggestion. Replacing "However" with "In contrast" in line 171 can help provide a clearer contrast between the two mentioned scenarios. I have made modifications to line 186 of the article. 5. The reviewer’s comment: Line 260: Please, remove “Additionally” and start the sentence from “The release of local inflammatory…” Response: Thank you for your kind suggestion. I have made modifications to lines 279 to 280 of the article. 6. The reviewer’s comment: Line 275: From “MnO2” and “Fe3O4”, put the numbers in subscripts. Apply this change in the subsequent lines. Response: Thank you for your kind suggestion. To properly represent the chemical formulas "MnO2" and "Fe3O4" with the numbers in subscripts, I have made the modifications in the article as requested. Thank you once again for your thorough review. 7. The reviewer’s comment: Line 541: Amend citation style (e.g., Renton et al. [173]). Response: Thank you for your kind suggestion. I have made modifications to line 562 of the article.

Reviewer 3 Report (Previous Reviewer 2)

General comments

I am deeply grateful to the authors for attending to my previous comments, which undoubtedly helped to improve what I consider to be a very innovative study. I will leave some minor details that I consider need to be corrected before publication.

Particular comments

Line 37. I consider that this term is a central point in your study, and it should be defined beforehand so that the reader understands your study from a conceptual point of view. I suggest to add it before the individualized terms.

Response:

Line 51. Please remove “In 2015”.

Response:

Line 71. Remove “In the 21st century”

Response:

Line 93. I suggest a more in-depth description of what the authors intend to convey with their figure, i.e., they could describe the differences between the different nanoformulations and their possible application in different areas.

Response:

Line 104. Please, add a reference.

Response:

Line 129. I appreciate your reference to the observation made above about both the therapeutic and toxic effects of these molecules. I suggest adding a brief discussion of the lack of information on the reduction of adverse effects, even if it is taken up in your perspectives.

Response:

Line 131. In addition to my previous comment, I suggest a brief discussion on the need for further research on the presence of adverse effects of the use of these substances or the therapeutic range they present.

Response:

Line 146- 151. This idea is very interesting, and I consider that they are contrasting statements. Therefore, I consider that it should be discussed, and it can be mentioned that "The application of nanoparticles has managed to expand for the administration of inhalation anesthetics as shown in a study done in dogs .... the results of these authors show that with the use of this innovative technology, it is possible to maintain general anesthesia in the animal biological model that would possibly have a benefit in reducing adverse effects, however ....."

Response:

Line 173. These results are interesting. I would encourage the authors to briefly discuss these findings and provide a biological explanation about it, possibly this could add more information about its application.

Response:

Line 275. Please remove the term "etc" and replace it with drugs such as gabapentin, amantadine, and cannabinoids.

Response:

Line 313. I consider this statement to be very important and would recommend the authors not only limit themselves to the fact that a benefit on the use of these particles may prolong the analgesic effect but might also increase the effectiveness of the analgesics. If possible, add this information to the discussion.

Response.

Line 332. Other factors or common causes of failure such as lack of training or lack of necessary equipment could also be mentioned.

Response:

Line 351. Please, correct the P value to P<0.05.

Response:

Line 377. I suggest replacing the term "is the science" with "the branch of pharmacology that uses...."

Response:

Author Response

1. The reviewer’s comment: Line 37. I consider that this term is a central point in your study, and it should be defined beforehand so that the reader understands your study from a conceptual point of view. I suggest to add it before the individualized terms. Response: Thank you for your suggestion. It is indeed important to provide a clear definition of the central term to ensure the reader's understanding of the study from a conceptual point of view. I have made modifications to lines 36 to 38 of the article. 2. The reviewer’s comment: Line 51. Please remove “In 2015”. Response: Thank you for your suggestion. I have removed "In 2015" from line 51. 3. The reviewer’s comment: Line 71. Remove “In the 21st century” Response: Thank you for your suggestion. I have removed "In the 21st century" from line 82. 4. The reviewer’s comment: Line 93. I suggest a more in-depth description of what the authors intend to convey with their figure, i.e., they could describe the differences between the different nanoformulations and their possible application in different areas. Response: Thank you for your suggestion. Providing a more in-depth description of the figure in line 93 can help readers better understand the differences between different nanoformulations and their potential applications. I have made modifications to lines 98 to 102 of the article. 5. The reviewer’s comment: Line 104. Please, add a reference. Response: Thank you for your kind suggestions. I have added the reference (doi:10.2174/1389200220666190610155049)to the article. Thank you again. 6. The reviewer’s comment: Line 129. I appreciate your reference to the observation made above about both the therapeutic and toxic effects of these molecules. I suggest adding a brief discussion of the lack of information on the reduction of adverse effects, even if it is taken up in your perspectives. Response: Thank you for your kind suggestion. I have made additions to lines 134 to 138 of the revised article. Thank you once again for your assistance. 7. The reviewer’s comment: Line 131. In addition to my previous comment, I suggest a brief discussion on the need for further research on the presence of adverse effects of the use of these substances or the therapeutic range they present. Response: Thank you for your suggestion. Adding a brief discussion on the need for further research regarding the presence of adverse effects or the therapeutic range of these substances in line 131 can enhance the completeness of the statement. I have made additions to lines 138 to 141 of the revised article. 8. The reviewer’s comment: Line 146- 151. This idea is very interesting, and I consider that they are contrasting statements. Therefore, I consider that it should be discussed, and it can be mentioned that "The application of nanoparticles has managed to expand for the administration of inhalation anesthetics as shown in a study done in dogs .... the results of these authors show that with the use of this innovative technology, it is possible to maintain general anesthesia in the animal biological model that would possibly have a benefit in reducing adverse effects, however ....." Response: Thank you for your suggestion. Adding a discussion of the contrasting statements and referencing a study to support the application of nanoparticles in the administration of inhalation anesthetics can provide valuable insights to the readers. I have made additions to lines 158 to 166 of the revised article. 9. The reviewer’s comment: Line 173. These results are interesting. I would encourage the authors to briefly discuss these findings and provide a biological explanation about it, possibly this could add more information about its application. Response: Thank you for your suggestion. Discussing the interesting findings in line 173 and providing a brief biological explanation can indeed enhance the understanding and relevance of the results. I have made additions to lines 188 to 192 of the revised article. 10. The reviewer’s comment: Line 275. Please remove the term "etc" and replace it with drugs such as gabapentin, amantadine, and cannabinoids. Response: Thank you for your suggestion. I have removed the term "etc" from line 275 and replaced it with specific examples of drugs. I have made additions to lines 294 to 295 of the revised article. 11. The reviewer’s comment: Line 313. I consider this statement to be very important and would recommend the authors not only limit themselves to the fact that a benefit on the use of these particles may prolong the analgesic effect but might also increase the effectiveness of the analgesics. If possible, add this information to the discussion. Response: Thank you for your suggestion. Expanding the discussion in line 313 to highlight not only the potential for prolonging the analgesic effect but also increasing the effectiveness of analgesics can provide a more comprehensive perspective. I have made additions to line 333 of the revised article. 12. The reviewer’s comment: Line 332. Other factors or common causes of failure such as lack of training or lack of necessary equipment could also be mentioned. Response: Thank you for your suggestion. Including other factors or common causes of failure, such as lack of training or lack of necessary equipment, in line 332 can provide a more comprehensive understanding of potential reasons for failure. I have made additions to lines 352 to 353 of the revised article. 13. The reviewer’s comment: Line 351. Please, correct the P value to P

This manuscript is a resubmission of an earlier submission. The following is a list of the peer review reports and author responses from that submission.

Round 1

Reviewer 1 Report

With pleasure, I read the review article titled "Nanotechnology, 3D printing technology, and genomics technology: Precision anesthesia supported by advanced technologies". Overall, the article reads very well and flows smoothly. The topic is clinically significant and if interest to the readership of Pharmaceutics. The authors provide a succinct introduction followed by relatively comprehensive summary of the existing literature. The materials are summarized in pertinent figures and tables. At the end of the review, the authors highlighted the limitations and future directions. I congratulate the authors for a well-done piece of literature. I have the following minor suggestions:

1. The authors need to clearly highlight the significance of their research. Is this the first ever systematic review on the topic? Have previous reviews been previously published? If so, what is new and special about the current manuscript? This is critical to establish the novelty of your work.

2. The authors claimed to “systematically” review the existing literature, which is not true. What has been presented is actually a narrative review of the literature, and this should be somehow added to the title of the article.

3. The methods would have been stronger if the authors opted to literally perform a “systematic review” by adhering to robust data search, selection, tabulation, and analysis. For example, the authors may want to tell which specific databases were searched, from when to when, what are the key words, what are the primary objectives, etc. However, I would just the reviewers to omit the claim of systematic by omitting the word.

4. Toward the end of the manuscript, to complement the limitations, the authors equally need to highlight the strengths of their report.

Reviewer 2 Report

General comments

I appreciate the opportunity to review the present article. I consider it very interesting because it reflects the current trends in the application of intelligent technology to improve anesthesia safety. This manuscript has the quality for publication. However, a weakness that I found was the lack of a clear objective so the reader can easily understand the aim of the study. 

Response:

Particular comments

Line 20. This sentence could be taken as a benefit obtained with the implementation of this technology. For example, "the implementation of these technologies has helped to improve safety during anesthesia and the effectiveness of analgesics....".

Response:

Line 21. In complement to my previous comment, I suggest that after your previous statement, you clearly mention the aim of this review. The topics that will be addressed are mentioned; however, will these topics be discussed or analyzed for any specific reason? As a recommendation, it could be mentioned that "the objective of this manuscript is to analyze the recent scientific evidence on the application of nanotechnology in anesthesiology .....".

Response:

Line 25. I suggest replacing the keyword "advanced technology" with "algology" and adding "anesthetic safety".

Response:

Line 37. Add a reference.

Response:

Line 42. This sentence needs more references to back up the statement.

Response:

Line 44. While this sentence may be clear to an experienced reader, I suggest adding a couple of examples of these consequences. For example, perhaps one of them is the clinical recognition of pain, where its evaluation often depends on the evaluator's experience in identifying pain-related clinical signs. I suggest adding a couple of examples that may help to position the relevance of the implementation of this technology.

Response:

Line 53. Similar to the observation made in the abstract, I suggest adding an objective to your systemic review.

Response:

Line 55. I consider this topic to be isolated from the other topics because it deals with an idea that is too general. I recommend synthesizing the conceptual aspects and deleting lines 56 - 60, then the rest could be moved to the subsequent topic starting on line 76.

Response: 

Lines 57- 62. As a complement to my previous comment, consider deleting these lines because it does not generate discussion with the main topic of your review. As an alternative, the authors could define nanotechnology and discuss its difference from nanomedicine, as mentioned in the name of the subtopic, and its application in medicine.

Response:

Line 74. Very good figure. To improve its impact on the text, I suggest expanding the description of the figure caption. For example, they could mention how the prolonged release of the drug is promoted due to the absorption mechanisms or by the chemical characteristics of the drug itself.

Response:

Line 88. Can all types of drugs be used with this technology? For example, some lipid-soluble drugs are used in patches to allow a prolonged release, does this concept apply with this technology? Please, discuss this.

Response:

Line 89. As a general recommendation to the authors, from a pharmacological perspective, the term "local anesthesia" is misused because it refers to the use of a drug that will promote the loss of consciousness, which is not the case. I suggest replacing this term with "local analgesia" to depict a meaning most related to its mechanism of action.

Response:

Line 91. This table is very interesting, and it accurately reflects what is implied in the text. Please, consider improving the table by adding pharmacokinetic parameters (if available) such as bioavailability, half-life, and time to therapeutic effect.

Response:

Line 98. I agree with this statement. I suggest including "which limits the scope of drug administration and clinical treatment effects because it depends entirely on the mechanisms of biotransformation and elimination of the organism so that the drug can exert its therapeutic effect. However, this itself can cause the prolongation of the adverse effects". Adding a couple of references would also be advised.

Response:

Line 99. "in vivo" and "in vitro" should be in italics. Apply this change throughout the text.

Response:

Line 102. If you mention that different authors support this, I expect to see two or three studies that support your idea. Please, add references.

Response:

Line 109. Could the authors mention if in this study they found any reduction of adverse effects? If not, I suggest mentioning if this could be an advantage due to the depressant effect on the cardiovascular system.

Response:

Line 124. Please include the dose range to observe toxic effects. For example, in cats, lidocaine has a short therapeutic range of 1 - 2 mg/ kg, but at 2.5 mg/kg, toxic effects such as methemoglobinemia can be observed. This technology could be an advantage for this species. Was this a reason for the implementation of this technology? Please, explain. 

Response:

Line 125. Add a reference.

Response:

Line 131. Could you mention the exact time range that increased the use of these molecules?

Response:

Line 216. This statement is very important. If the authors allow me, I suggest that it can be rephrased with "this event can lead to activation of peripheral sensitization phenomena, which can lead to hyperalgesia and allodynia". The following references might be helpful:  

Bell (2019) The neurobiology of acute pain The Veterinary Journal doi: https://doi.org/10.1016/j.tvjl.2018.05.004, Ellison (2017) Physiology of pain Critical Care Nursing Clinics of North American Doi: https://doi.org/10.1016/j.cnc.2017.08.001, Domínguez Oliva (2023) The neurobiology of pain and facial movements in rodents: Clinical applications and current research Doi: https://doi.org/10.3389/fvets.2022.1016720.

Response: 

Line 219. This sentence is very important. Here, the authors could add that it is worth mentioning that despite the therapeutic importance of these drugs, they may present a paradoxical effect of generating allodynia phenomena due to the constant use or overdosage of these drugs.

Response:

Line 221. Please, add references.

Response:

Line 223. Please add a reference.

Response:

Line 227. Could you please add the value of r obtained in the study and if this was significant?

Response:

Line 229. Please, remove the year. And amend the in-text citation style throughout the manuscript.

Response.

Line 230. Please, add which was the study population, i.e., it was patients who were admitted to intensive care or who were recovering from anesthesia?

Response:

Line 251. Could the authors mention the incidence of failure of epidural analgesic administration? Since this is another limitation of the technique.

Response:

Lines 261- 263. Consider improving the statement in these lines. A suggestion could be: "Multimodal analgesia is defined as the use of two or more analgesics of different classes to inhibit pain in different sites of the neurobiology of pain, such concept allows to improve pain management". To complement this idea, I encourage the authors to consult the following articles Hernández Ávalos (2021) Nociceptive pain and anxiety in equine: Physiological and behavioral alterations Doi: www.doi.org/10.14202/vetworld.2021.2984-2995. Helander (2017) Multimodal Analgesia, Current Concepts, and Acute Pain Considerations. Curr Pain Headache Rep 21, 3 (2017). https://doi.org/10.1007/s11916-017-0607-y.

Response:

Line 265. Please add "/" to the term anesthetic analgesia.

Response:

Line 267. How did the authors determine the efficiency of the system? Could you give more details of the experiment? Please

Response:

Line 272. Please, add more references.

Response:

Line 286. Could the authors provide a prior definition of pharmacogenomic?

Response:

Line 312. Please, define the abbreviation "POR".

Response:

Lines 316 – 318. This section is isolated, and I consider that it should be expanded because it is directly related to its subject. In addition, add references about the drugs (e.g., dissociative anesthetics or alpha 2 adrenergic drugs).

Response:

Line 333. Please add a reference.

Response:

Line 340. Add a reference.

Response:

Line 537. I recommend adding some conclusions according to the aim of the study.

Response:

Reviewer 3 Report

The present article is a highly relevant topic in medicine, not only in anesthesia but in several medical areas. The application of innovative technologies provides important benefits for human health. For example, nanotechnology is applied to drug administration to reduce the adverse effects of certain drugs. Likewise, genomics can help to target some medical conditions or drugs’ mechanisms of action through specific receptors or enzymes. The overall review that this study presents highlights the role of precision medicine in current worldwide issues. I left some comments hoping they can be helpful for the authors. 

Title: Please, consider if it would be appropriate to modify the order of the technologies mentioned in the title, changing it from “Nanotechnology, 3D printing technology, and genomics technology: Precision anesthesia supported by advanced technologies” to “Nanotechnology, genomics, and 3D printing technology: Precision anesthesia supported by advanced technologies”. This suggestion is because the order of the heading and subheadings inside the text follow the suggested order (2. Nanotechnology, 3. Genomics, 4. 3D printing).

Response:

Lines 29-31: To emphasize the importance of precision medicine, I would recommend adding some brief examples of the current application of precision medicine in different biomedical fields or into the main public health concerns. For example, in cancer research, neurodegenerative diseases such as Alzheimer’s, heart diseases, anesthesia (the main topic of the article), and even COVID-19 where precision treatment has been studied.

Response: 

 Lines 36-44. This paragraph needs some references.

Response:

Lines 51-52:  The aim of the article states that “This article systematically reviews…”. When reading this sentence, I expected to see the search methodology according to PRISMA guidelines and more information about the criteria used to search and select the 135 studies included in the review. In my opinion, there are two options: a) to rephrase the aim of the study or b) to include the search methodology.

Response:

Figure 1 is a very good visual summary of the current nanocarriers used in medicine.

Response:

Lines 85-88: I recommend specifying if the enhanced solubility, dissolution kinetics, drug concentrations, etc., is referring to local anesthetics, general anesthetics, or inhalant drugs (or all).

Response:

Figure 2. Could the authors increase the figure size or increase the font size? As it is, the font size is too small and blurry.

Response:

Line 135: Consider starting a new paragraph from “Similarly, other animal studies…”

Response:   

Line 202: Start a new paragraph from “Ophthalmic anesthesia…”

Response:

Lines 203-207: Please, add the name of the drugs that have been administered as nano-microemulsions or nano-emulsions for ophthalmic anesthesia.

Response:

Lines 216-217: Please, add that this phenomenon is called “allodynia”.

Response:

Lines 245-248: I recommend mentioning in which types of surgical procedures epidural catheter is used for pain management.

Response:

Lines 261-262: It needs to be added that multimodal analgesia consists in two or more analgesic drugs with different mechanisms of action, so they can act at different levels of the pain recognition pathway.

Response:  

Line 537: I suggest adding a “Conclusions” section to summarize the main findings according to the revised literature.

Response:

Reviewer 4 Report

In clinical anesthesiology, it is essential to monitor the patient's vital signs, in order for him to remain stable; In addition, another of the objectives of the anesthesiologist is to resolve emergencies or complications that may be observed during the perioperative period, including maintaining an appropriate depth of anesthesia, correct sedation, analgesia, amnesia, muscle relaxation, autonomous protection, as well as the reduction of stress reactions. However, with technological advances, anesthesiologists must not only master basic anesthesia procedures but also take advantage of the convenience afforded by advanced science and technology to improve such techniques. Consequently, nanomedicine may allow the advancement and optimization of clinical anesthesiology practice in perioperative settings. The main strength of this review manuscript is that it provides a systematic review of current developments in nanotechnology, genomic technology, and 3D printing technology with respect to clinical anesthesia, thereby exploring the possibility of achieving more precise anesthesia.

Nevertheless, some points must be addressed to achieve publication quality. I have left some comments hoping that they can help the authors.

General comments

L40: please add a reference

L44: please add a reference

L53: After the introduction, add a section called methodology. In it, indicate the sites or sources used to search for information, keywords used in the search, inclusion and exclusion criteria taken into account, or even how you classified the articles consulted according to their content. If possible, support your explanation with a figure.

L77-79: please change the wording of these lines to; including maintaining an appropriate depth of anesthesia, sedation, amnesia and analgesia, muscle relaxation, neurovegetative protection, and reducing stress reactions.

L82: please add a reference

Table 1: I suggest to the authors the analysis of the following articles so that the possibility of integrating them into the table in the corresponding sections:

10.2460/javma.256.9.1011

10.2460/ajvr.22.01.0007

L120: Regarding inhalational anesthetics, please discuss and add to your manuscript the following papers:

10.1016/j.vaa.2016.02.004

10.1080/21691401.2019.1624557

L125: please add a reference

L211: Liposomes have not only been added to contact lenses; they have also been added to retinal neuroprotection therapies. This citation refers to this fact, therefore I suggest to the authors may be considered:

10.1021/acsnano.8b00596

L219: please add a reference

L224: please add a reference

L226: please add more discussion on current pain management strategies with the help of nanotechnology knowledge, for this, I leave a reference that I hope can help the authors:

10.1080/21691401.2018.1553885

L277: I suggest authors discuss the use of nanocarriers, nanoparticles, NSAIDs, MgO, MnO2, Fe3O4, or other analgesics for pain management through nanotechnology in more depth. I wish you could consider the following references:

10.1016/j.tips.2021.03.007

10.1155/2014/394264

10.2174/1381612821666151027152752

10.3390/moléculas27217312

10.1080/21691401.2017.1313265

10.1186/s12951-022-01473-y

L284: please add a reference

L293: please add a reference

Table 2: Other related genes and variants have been identified, particularly with tramadol. I leave a reference that I hope can help the authors:

10.5455/javar.2021.h529

L318: please clarify, which anesthetics.

L436: Other drugs that should be considered due to the neuromuscular blocks that they can generate and that some authors conceptualize as serious complications are antibiotics such as polymyxins, lincosamides, and aminoglycosides. Please discuss these effects as well.

L537: please add a section where you write the conclusions of your manuscript.